# Structure-based design of chimeric antigens for multivalent protein vaccines

S. Hollingshead[1], I. Jongerius[1,2], R.M. Exley[1], S. Johnson[1], S.M. Lea [1] &  C.M. Tang[1]

There is an urgent need to develop vaccines against pathogenic bacteria. However, this is often hindered by antigenic diversity and difficulties encountered manufacturing membrane proteins. Here we show how to use structure-based design to develop chimeric antigens (ChAs) for subunit vaccines. ChAs are generated against serogroup B *Neisseria meningitidis* (MenB), the predominant cause of meningococcal disease in wealthy countries. MenB ChAs exploit factor H binding protein (fHbp) as a molecular scaffold to display the immunogenic VR2 epitope from the integral membrane protein PorA. Structural analyses demonstrate fHbp is correctly folded and the PorA VR2 epitope adopts an immunogenic conformation. In mice, immunisation with ChAs generates fHbp and PorA antibodies that recognise the antigens expressed by clinical MenB isolates; these antibody responses correlate with protection against meningococcal disease. Application of ChAs is therefore a potentially powerful approach to develop multivalent subunit vaccines, which can be tailored to circumvent pathogen diversity.

[1] Sir William Dunn School of Pathology, University of Oxford, South Parks Road, Oxford OX1 3RE, UK. [2]Present address: Department of Medical Microbiology, University Medical Centre Utrecht, Heidelberglaan 100, 3584 CX Utrecht, Netherlands. Correspondence and requests for materials should be addressed to C.M.T. (email: christoph.tang@path.ox.ac.uk)

oxoid and capsule-based vaccines have prevented millions of deaths caused by bacterial pathogens. Toxoid-based vaccines have almost eliminated diphtheria and tetanus in wealthy countries[1], while capsule-based vaccines have substantially reduced disease caused by *Haemophilus influenzae*[2], *Streptococcus pneumoniae*[3], and some strains of *Neisseria meningitidis*[4]. However, challenges remain in developing vaccines against pathogens for which toxoid and capsule-based vaccines are not feasible. These pathogens include non-typeable strains of *H. influenzae* and *S. pneumoniae*[2,3], un-encapsulated pathogens such as *Neisseria gonorrhoeae* and *Moraxella catarrhalis*[5,6] and encapsulated serogroup B *N. meningitidis*, for which a capsule-based vaccine is not feasible[7]. Given the rise in the emergence of multi-drug resistant bacteria[8], new approaches for vaccine development are required. However, strategies for generating successful vaccines are hampered by pathogen diversity[9] and the difficulties associated with presenting epitopes from membrane-embedded surface proteins to the immune system[10].

Two main approaches have been used to develop vaccines against serogroup B *N. meningitidis*; outer membrane vesicle vaccines (OMVV) and recombinant protein subunit vaccines. OMVVs were first developed in the 1980s[11–13]. The immunodominant antigen in meningococcal OMVVs is PorA[14], an abundant outer-membrane porin with eight surface-exposed loops[15]. Loops one and four are termed variable region 1 and 2 (VR1 and VR2), respectively, as they generate immune responses and are subject to antigenic variation[16]. The VR2 loop dominates PorA-specific immunity elicited by OMVVs, which offer limited or no cross-protection against strains expressing PorA with a different VR2[17,18]. To broaden coverage, OMVVs containing multiple PorAs have been developed[19–21] and selected for the prevalence of PorA sequences in circulating strains[21,22]. However, OMVVs present complex manufacturing and regulatory issues[23]. The recombinant subunit vaccines Bexsero and Trumenba contain an important meningococcal antigen, factor H binding protein (fHbp), which is a lipoprotein composed of two β-barrels that tightly bind domains 6 and 7 of human complement factor H (CFH)[24–27]. fHbp is antigenically variable; databases of genome sequences contain more than 900 different fHbp peptides[28], which fall into three variant groups or two subfamilies: V1 (subfamily B), V2 and V3 (both subfamily A)[29,30]. In general, immunisation with a particular fHbp induces cross-protection against strains that express fHbp belonging to the same, but not a different, variant group, although there can be cross-protection between fHbp variant groups 2 and 3 (subfamily A)[31,32]. Bexsero contains a single fHbp peptide (V1.1), with two other recombinant antigens as well as an OMV[33], while Trumenba is composed solely of two fHbp peptides (V1.55 and V3.45)[34]. Antigens in Bexsero and Trumenba have exact sequence matches to 36 and 4.8% of serogroup B *N. meningitidis* disease isolates currently circulating in the UK, respectively[28], leading to concerns about their ability to provide broad coverage against an antigenically diverse pathogen.

Here we use structure-based design to generate chimeric antigens (ChAs) against serogroup B *N. meningitidis*. ChAs exploit fHbp as a molecular scaffold to present the surface-exposed PorA VR2 loop, which is achieved by inserting the VR2 loop into a β-turn region in fHbp. ChAs retain epitopes from both fHbp and PorA, and can elicit functional immune responses against both antigens. We demonstrate that integration of a VR2 loop does not alter the overall architecture of fHbp and that the VR2 loop folds into a conformation recognised by a bactericidal mAb. We provide proof-in-principle that ChAs can be used to display selected epitopes from integral membrane proteins, such as PorA.

## Results

**Design and construction of ChAs.** The surface-exposed proteins fHbp and PorA (Fig. 1a) elicit bactericidal antibody responses, which are a correlate of protection against meningococcal disease[34]. fHbp is a lipoprotein that can be expressed as a soluble protein in *Escherichia coli* following removal of the N-terminal lipobox motif[32]. As an isolated extracellular loop, the PorA VR2 is likely soluble when expressed separately from the hydrophobic β-barrel of PorA. We exploited soluble fHbp peptide V1.1 as a molecular scaffold to display the PorA VR2 loop, P1.16 (VR2 P1.16). VR2 P1.16 (YYTKDTNNNLTLVP) was inserted into six different β-turn regions in fHbp. At each VR2 P1.16 insertion site, a single amino acid was deleted from the fHbp scaffold (Fig. 1b). The resulting ChAs were named according to a scheme whereby fHbp$^{V1.1}$:PorA$^{151/P1.16}$ denotes fHbp V1.1 with VR2 P1.16 inserted in the place of fHbp residue 151 (Supplementary Table 1). ChAs all express to high levels in *E. coli* and are purified by nickel affinity chromatography. Western blot analyses confirm all ChAs retain epitopes recognised by an α-P1.16 mAb and α-fHbp pAbs (Fig. 1c).

**ChAs are stable and can bind CFH.** Stability of an antigen is an important property of a vaccine, and insertion of PorA epitopes might disrupt the overall structure of the ChA scaffold. Therefore, we determined the thermal stability of ChAs by differential scanning calorimetry (DSC, Table 1 and Supplementary Figure 1A). Insertion of VR2 P1.16 into the N- or C-terminal β-barrel of fHbp decreased the thermal stability of that β-barrel by 1.0–15.5 °C, with little or no effect on the other β-barrel. Overall, the lowest measured melting temperature ($T_m$) of any β-barrel was 60.5 °C, which is considerably higher than the N-terminal $T_m$ of V3.45 (41 °C), one of the fHbp peptides in Trumenba® (Table 1).

A key property of fHbp is its ability to bind CFH[24–26] (Supplementary Figure 2A). Therefore, surface plasmon resonance (SPR) was used to determine the affinity of each ChA for domains 6 and 7 of CFH (CFH$_{6/7}$, Table 1 and Supplementary Figure 3). Similar to wild-type fHbp, most ChAs bind CFH$_{6/7}$ at high affinity, indicating the fHbp scaffold retains its function. The exceptions were fHbp$^{V1.1}$:PorA$^{183/P1.16}$, to which there was no detectable CFH$_{6/7}$ binding, and fHbp$^{V1.1}$:PorA$^{267/P1.16}$, to which CFH$_{6/7}$ binding was reduced by approximately eight-fold. In these two ChAs, VR2 P1.16 is situated in the fHbp/CFH interface (Supplementary Figure 2B), potentially inhibiting CFH$_{6/7}$ binding.

**fHbp:PorA ChAs generate protective immune responses.** To examine the ability of ChAs to elicit immune responses, we immunised groups of CD1 mice with each ChA using alum or monophospholipid A (MPLA) as the adjuvant (Fig. 2a). Post immunisation, sera were obtained from mice and pooled. Immune responses were assessed against *N. meningitidis* H44/76, a serogroup B strain which expresses fHbp V1.1 and PorA VR2 P1.16. We constructed isogenic strains to define immune responses directed against fHbp (H44/76Δ*porA*) and PorA (H44/76Δ*fHbp*), as well as a H44/76Δ*fHbp*Δ*porA* control. Western blot analyses of lysates from these strains demonstrate that all ChAs elicited antibodies that recognise both fHbp and PorA (Fig. 2b, d), with the exception of fHbp$^{V1.1}$:PorA$^{267/P1.16}$/MPLA, which generated sera that only recognised fHbp (Fig. 2d).

Next, we used flow cytometry to assess the ability of antibodies raised by immunisation with ChAs to recognise fHbp and PorA on the surface of *N. meningitidis*. We detected antibody binding to fHbp and PorA by flow cytometry, using *N. meningitidis* strains H44/76Δ*porA* and H44/76Δ*fHbp*, respectively. Results

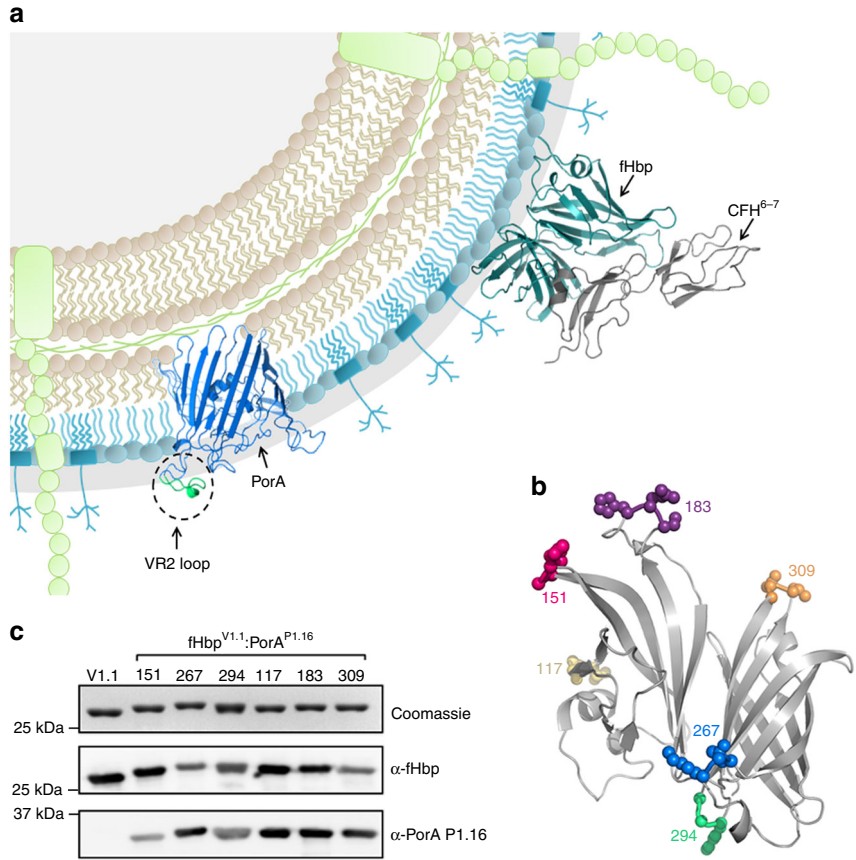

**Fig. 1** Structure-based design of ChAs. **a** Schematic of meningococcal cell surface, depicting the key surface-exposed antigens, fHbp and PorA. **b** Location of fHbp residues replaced with VR2 P1.16. **c** Analysis of recombinant ChAs by SDS-PAGE and western blot. Immunoblots are probed with α-V1.1 fHbp pAb and α-PorA P1.16 mAb. Complete gel and western blots are shown in Supplementary Figure 5

**Table 1 ChA stability and affinity for CFH$_{6/7}$**

| Protein | Melting temperature (°C) | | Affinity for CFH6/7 |
|---|---|---|---|
| | N-terminal | C-Terminal | $K_d$ (nM) |
| fHbp V1.1 | 69.8 | 87.9 | 3.6 ± 0.4 |
| fHbp$^{V1.1}$:PorA$^{151/P1.16}$ | 60.5 | 87.3 | 2.3 ± 0.2 |
| fHbp$^{V1.1}$:PorA$^{267/P1.16}$ | 68.4 | 75.7 | 28 ± 3.0 |
| fHbp$^{V1.1}$:PorA$^{294/P1.16}$ | 61.7 | 72.4 | 2.7 ± 0.4 |
| fHbp$^{V1.1}$:PorA$^{117/P1.16}$ | 65.0 | 87.9 | 2.3 ± 0.8 |
| fHbp$^{V1.1}$:PorA$^{183/P1.16}$ | 68.8 | 88.0 | NBD |
| fHbp$^{V1.1}$:PorA$^{309/P1.16}$ | 69.0 | 76.9 | 5.5 ± 0.6 |
| fHbp V1.4 | 64.4 | 89.4 | – |
| fHbp V3.45 | 41.0 | 83.0 | – |
| fHbp$^{V1.4}$:PorA$^{151/P1.1.10\_1}$ | 54.0 | 89.0 | – |
| fHbp$^{V1.4}$:PorA$^{151/P1.14}$ | 55.0 | 88.0 | – |
| fHbp$^{V1.4}$:PorA$^{151/P1.15}$ | 55.0 | 89.0 | – |
| fHbp$^{V3.45}$:PorA$^{158/P1.4}$ | 40.0 | 81.0 | – |
| fHbp$^{V3.45}$:PorA$^{158/P1.9}$ | 39.0 | 80.0 | – |

$K_d$ dissociation constant of binding to CFH, *NDB* no binding detected

were compared with findings obtained with control sera from mice immunised with phosphate-buffered saline (PBS) and adjuvant alone. Antisera raised against each ChA detected fHbp on the bacterial surface ($p \leq 0.0001$, Figs. 2c, e), and certain antisera also demonstrated significant binding to PorA: antisera from mice immunised with alum and ChAs fHbp$^{V1.1}$:PorA$^{294/}$ P1.16, fHbp$^{V1.1}$:PorA$^{117/P1.16}$, fHbp$^{V1.1}$:PorA$^{183/P1.16}$ or fHbp$^{V1.1}$:

PorA$^{309/P1.16}$ ($p \leq 0.01$ by two-way ANOVA, Fig. 2c), and from mice immunised with MPLA and ChAs fHbp$^{V1.1}$:PorA$^{151/P1.16}$, fHbp$^{V1.1}$:PorA$^{294/P1.16}$ or fHbp$^{V1.1}$:PorA$^{309/P1.16}$ (Fig. 2e, $p \leq$ 0.05 by two-way ANOVA). Significant binding ($p \leq 0.05$ by two-way ANOVA) is observed with fHbp$^{V1.1}$:PorA$^{267/P1.16}$/MPLA antisera to the H44/76Δ$fHbp\Delta porA$ negative control strain, which is due to non-specific binding (Supplementary Figure 4).

The serum bactericidal assay (SBA) assesses the ability of antibodies to initiate complement-mediated lysis of *N. meningitidis*. When using baby rabbit complement, an SBA titre of $\geq 8$ is an accepted correlate of protective immunity against *N. meningitidis*[35]. SBAs conducted with each set of pooled ChA/adjuvant antisera and wild-type *N. meningitidis* H44/76 all had titres of $\geq 128$ (Fig. 3a). Significantly higher titres ($p \leq 0.05$ by two-way ANOVA) were observed for antisera raised against fHbp$^{V1.1}$:PorA$^{151/P1.16}$, fHbp$^{V1.1}$:PorA$^{117/P1.16}$, fHbp$^{V1.1}$: PorA$^{183/P1.16}$ and fHbp$^{V1.1}$:PorA$^{309/P1.16}$ when MPLA, rather than alum, was used as the adjuvant.

To measure fHbp-specific responses, we performed SBAs with *N. meningitidis* H44/76Δ$porA$; all sets of ChA/adjuvant antisera generated protective SBA titres $\geq 256$ (Fig. 3b). To evaluate α-PorA-specific responses we initially performed SBAs with pooled antisera; however, no PorA-dependent serum bactericidal activity was detected. Therefore, we examined PorA-dependent responses in individual mice. SBA titres were detected with antisera raised against fHbp$^{V1.1}$:PorA$^{294/P1.16}$ when alum was used as the adjuvant, and with antisera raised against fHbp$^{V1.1}$:PorA$^{151/}$ P1.16 (a low level of SBA from a single mouse), or fHbp$^{V1.1}$: PorA$^{309/P1.16}$ when MPLA was the adjuvant (Fig. 3c). Although PorA was detected on the surface of *N. meningitidis* by antisera

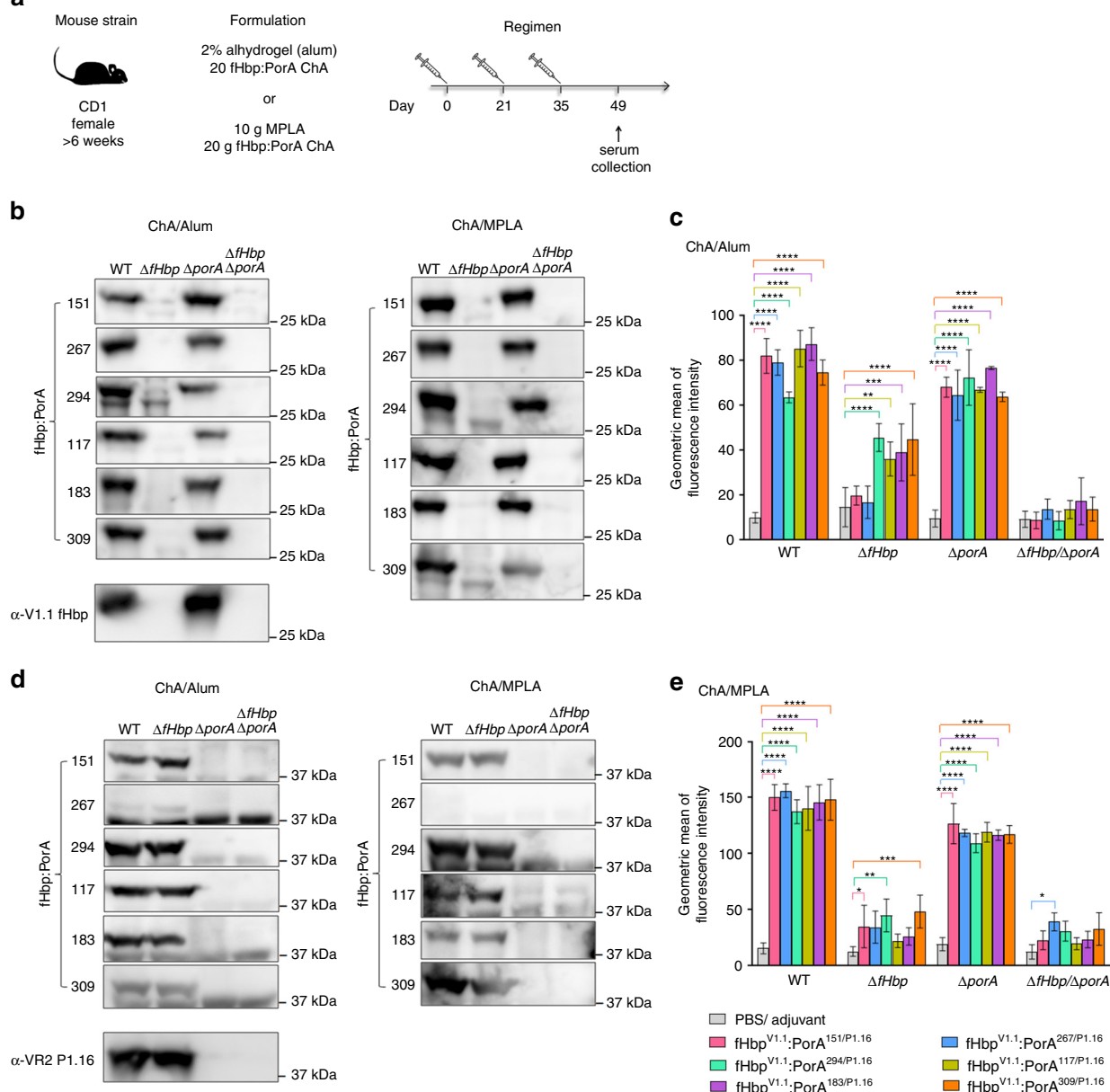

**Fig. 2** Detection of fHbp and PorA by ChA antisera. **a** Immunisation strategy. Mice ($n = 8$ per group) were subcutaneously immunised three times with each combination of ChA/Adjuvant. **b**, **d** Western blots of *N. meningitidis* whole-cell lysates probed with each set of ChA/adjuvant antisera: **b** detection of fHbp, **d** detection of PorA, complete western blots are in Supplementary Figures 6 and 7. **c**, **e** Flow cytometry analysis showing binding of each set of ChA/adjuvant antisera to *N. meningitidis* H44/76 strains WT, Δ*fHbp*, Δ*porA* and Δ*fHbp*Δ*porA*. s.d. of independent assays ($n = 3$) is indicated. Two-way ANOVA and Dunnett's method of multiple comparison were used to compare the fluorescence intensity of ChA antisera with PBS control sera (**c**, **e** *$p \leq 0.05$, **$p \leq 0.01$, ***$p \leq 0.001$, ****$p \leq 0.0001$)

from mice immunised with alum and fHbp^V1.1:PorA^117/P1.16 or fHbp^V1.1:PorA^183/P1.16, or with MPLA and fHbp^V1.1:PorA^294/P1.16 (Fig. 2c, e), these antigens/adjuvants did not elicit PorA-dependent SBA titres.

To activate the classical pathway, bound immunoglobulin (Ig) must recruit the C1q subunit of C1[36]. The ability of Ig classes to bind C1q varies; a single IgM can be sufficient for C1q recruitment[37], while several IgGs must be bound in close proximity and in a particular conformation[38]. Therefore, we examined which Ig isotypes are elicited by ChAs. Flow cytometry demonstrated that IgG1 was the predominant Ig bound to the surface of *N. meningitidis* (Fig. 3d–g). When compared with sera from mice immunised with PBS/adjuvant alone, all ChAs elicited significant α-fHbp-specific IgG1 responses (Fig. 3d, f, $p \leq 0.0001$ by two-way ANOVA). However, significant IgG1 binding to PorA on the surface of H44/76Δ*fHbp* was observed only with antisera raised against fHbp^V1.1:PorA^294/P1.16, fHbp^V1.1:PorA^117/P1.16, fHbp^V1.1:PorA^183/P1.16 or fHbp^V1.1:PorA^309/P1.16 with alum ($p \leq 0.01$ by two-way ANOVA; Fig. 3e), and fHbp^V1.1:PorA^151/P1.16 or fHbp^V1.1:PorA^294/P1.16 with MPLA ($p \leq 0.0001$ by two-way ANOVA; Fig. 3g). Interestingly, there was no detectable IgG1 binding to PorA using sera raised against fHbp^V1.1:PorA^309/P1.16/MPLA, against which two mice had α-PorA SBA titres (Fig. 3c); instead this antisera contained significant levels of α-PorA-specific IgG2a and IgM ($p \leq 0.01$ by two-way ANOVA; Fig. 3g).

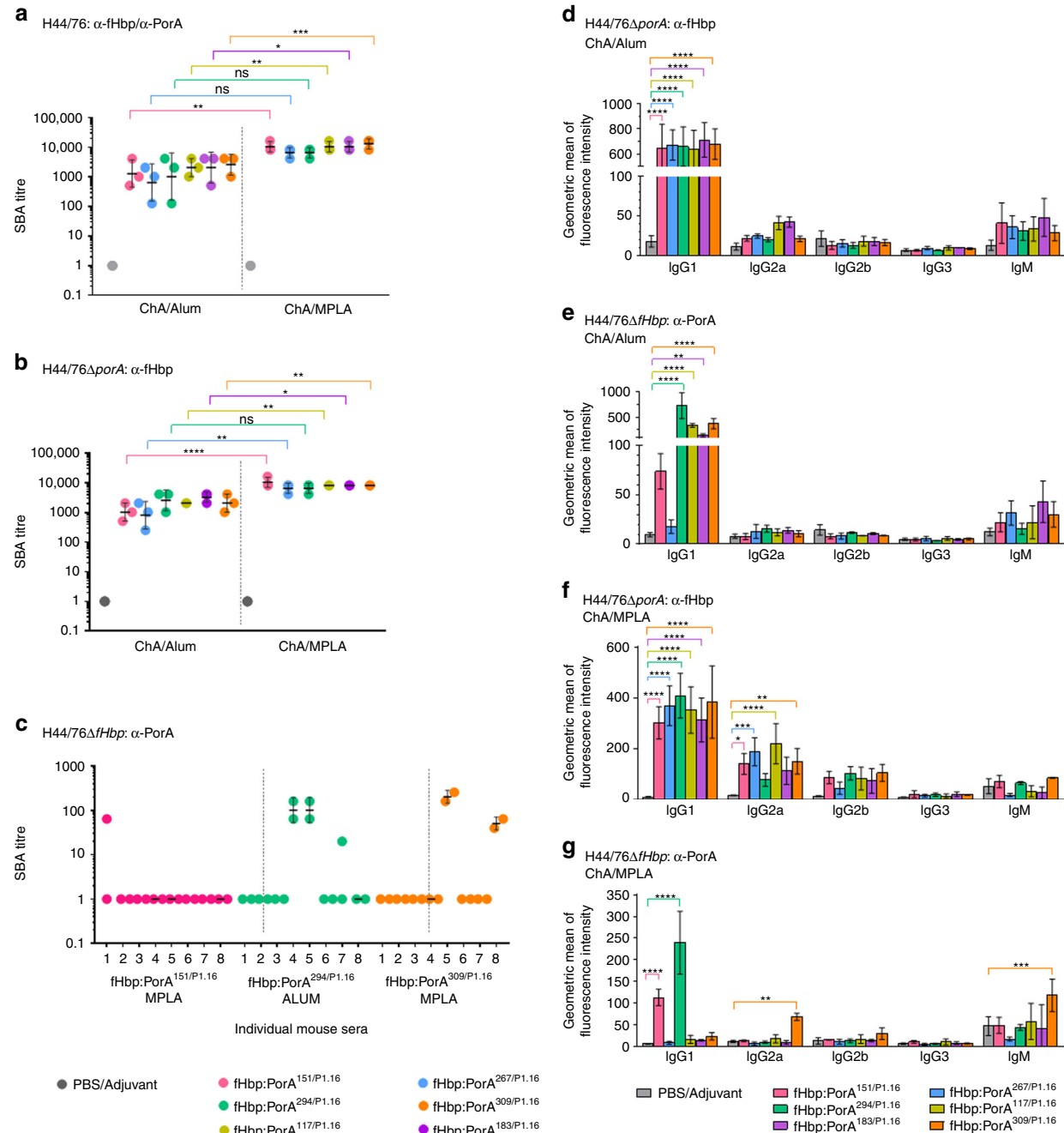

**Fig. 3** Immunogenicity of ChAs. Titres from SBAs performed with *N. meningitidis* strains H44/76 (**a**, *n* = 3), H44/76Δ*porA* (**b**, *n* = 3), and H44/76Δ*fHbp* (**c**, *n* = 2) and ChA/adjuvant antisera. SBAs with H44/76 and H44/76Δ*porA* were conducted using pooled ChA/adjuvant antisera, and SBAs with H44/76Δ*fHbp* were conducted with ChA/adjuvant antisera from individual mice. Geometric mean and s.d. of independent assays (*n* > 2) are indicated. Flow cytometry was used to detect binding of mouse isotypes IgG1, IgG2a, IgG2b, IgG3 and IgM in ChA antisera to *N. meningitidis* strains H44/76Δ*porA* (**d**, **f**) and H44/76Δ*fHbp* (**e**, **g**). Mouse isotypes binding to *N. meningitidis* were detected with isotype specific secondary antibodies; s.d. of independent assays (*n* = 3) is indicated. Two-way ANOVA and Dunnett's method of multiple comparison were used to compare SBA titres from pooled antisera (**a**, **b**) and ChA antisera to PBS control sera (**d**–**g**) (*$p \leq 0.05$, **$p \leq 0.01$, ***$p \leq 0.001$, ****$p \leq 0.0001$)

**ChAs retain the architecture of fHbp and the PorA loop.** We determined the atomic structures of ChAs with VR2 P1.16 inserted into position 151, 294, or 309, all of which generated PorA-dependent SBA titres. These structures were solved using molecular replacement to resolutions of 2.9 Å for fHbp$^{V1.4}$:PorA$^{151/P1.16}$, 3.7 Å for fHbp$^{V1.1}$:PorA$^{294/P1.16}$ and 2.6 Å for fHbp$^{V1.4}$:PorA$^{309/P1.16}$. All ChA scaffolds align with good agreement (route-mean-square deviations (RMSDs) range between 0.447 and 0.614) to fHbp V1.1 (Supplementary

Figure 2C), demonstrating that the fHbp scaffold is not perturbed by insertion of VR2 P1.16. In each structure VR2 P1.16 adopts a β-turn conformation that extends away from the fHbp scaffold (Fig. 4a). Importantly, these VR2 P1.16 conformations all mimic a VR2 P1.16 peptide in a complex with a bactericidal Fab fragment[39] (Fig. 4b, RMSDs of atomic positions range between 0.346 and 0.970), demonstrating that in ChAs VR2 P1.16 adopts a conformation that can induce bactericidal antibody responses[40].

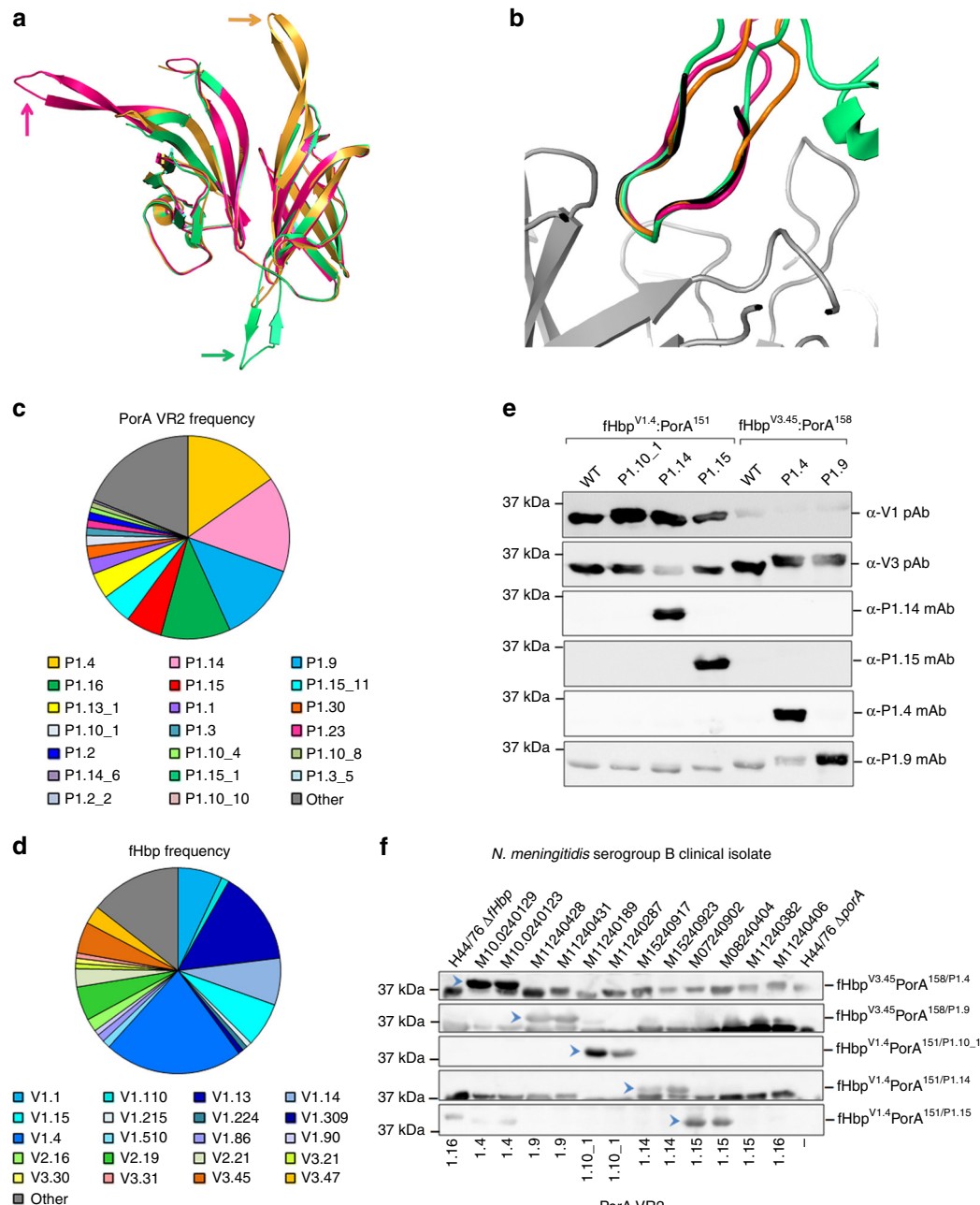

**Fig. 4** Structure of ChAs and frequency of fHbp/PorA alleles in the UK. **a** Alignment of fHbp scaffolds from fHbp$^{V1.4}$:PorA$^{151/P1.16}$ (pink), fHbp$^{V1.1}$:PorA$^{294/}$ $^{P1.16}$ (green) and fHbp$^{V1.4}$:PorA$^{309/P1.16}$ (orange), VR2 P1.16 is indicated for each ChA by the correspondingly coloured arrow. **b** Alignment of VR2 P1.16 region 'KDTNNNL' from fHbp$^{V1.4}$:PorA$^{151/P1.16}$ (pink), fHbp$^{V1.1}$:PorA$^{294/P1.16}$ (green) and fHbp$^{V1.4}$:PorA$^{309/P1.16}$ (orange) with the P1.16 peptide 'KDTNNNL' (black) in a complex with a bactericidal Fab fragment (grey) from mAb MN12H2 (PDB ID: 2MPA). Frequency of PorA VR2 (**c**) and fHbp peptides (**d**) in *N. meningitidis* serogroup B strains ($n = 243$) isolated in 2016 in the UK. Data downloaded from the Meningococcal Research Foundation, 27 June 2017 (ref. [41]). Other: remaining alleles that occur in <4 isolates. **e** Analysis of recombinant ChAs by SDS-PAGE and western blot. Immunoblots are probed with α-PorA VR2 mAbs: P1.4, P1.9, P1.14 and P1.15. **f** Detection of PorA (indicated by blue arrows) in a panel of *N. meningitidis* serogroup B isolates by mouse polyclonal antisera from ChAs fHbp$^{V1.4}$:PorA$^{151/P1.1.10\_1}$, fHbp$^{V1.4}$:PorA$^{151/P1.14}$, fHbp$^{V1.4}$:PorA$^{151/P1.15}$, fHbp$^{V3.45}$:PorA$^{158/P1.4}$ and fHbp$^{V3.45}$:PorA$^{158/P1.9}$. Complete western blots are shown in Supplementary Figures 8 and 9

**ChAs with different PorA VR2 loops generate protective responses**. To test the adaptability of ChAs, we generated several ChAs composed from different combinations of fHbp and PorA VR2. The comprehensive meningococcal genome data available for strains isolated in the UK enable design of ChAs that have exact sequence matches to the most common antigens in a given region[41]. In 2016, the most prevalent PorA VR2 peptides in serogroup B *N. meningitidis* isolates were P1.4 (15.2%), P1.14

(15.2%), P1.9 (12.8%), P1.16 (11.1%) and P1.15 (5.8%, Fig. 4c). While the most prevalent fHbp peptides belonging to variant groups 1 and 3 were V1.4 and V3.45, present in 21.8% and 4.9% of serogroup B *N. meningitidis* isolates, respectively (Fig. 4d). We constructed five different ChAs, in which a VR2 peptide was inserted into position 151 in V1.4 or position 158 in V3.45 (Fig. 1b). Following ChA expression and purification, western blot analyses confirmed these ChAs, all retained epitopes

**Table 2 Serum bactericidal assay titres**

| Pooled antisera | Serogroup B isolate | fHbp peptide | PorA VR2 | SBA titre |
|---|---|---|---|---|
| fHbp$^{V3.45}$:PorA$^{158}$/P1.4 | M10240123 | V1.92[a] | P1.4 | 1/160 |
| fHbp$^{V3.45}$:PorA$^{158}$/P1.9 | M11240431 | V2.19 | P1.9 | 1/1280 |
| fHbp$^{V1.4}$:PorA$^{151}$/P1.1.10_1 | M11240189 | V3.84 | P1.10_1 | 1/20 |
| fHbp$^{V1.4}$:PorA$^{151}$/P1.14 | M15240853 | V3.45 | P1.14 | 1/640 |

α-PorA SBA titres generated using pooled ChA/alum antisera and serogroup B *N. meningitidis* isolates with mismatched fHbp variants
[a]fHbp truncated at residue 242

recognised by their cognate α-VR2 mAb and α-fHbp pAbs (Fig. 4e).

To examine the ability of these ChAs to elicit immune responses, groups of CD1 mice were immunised with each ChA and alum (Fig. 2a); antisera obtained post immunisation were pooled. To assess the resulting PorA immune responses, western blots were conducted with pooled antisera and cell lysates from a panel of serogroup B *N. meningitidis* disease isolates. Figure 4f demonstrates that all ChAs elicited PorA-specific antibodies that recognised their cognate PorA VR2. To evaluate PorA-mediated SBA responses, we performed SBAs with pooled ChA/alum antisera and serogroup B *N. meningitidis* strains with mismatched fHbp variants to negate fHbp cross-protection. Titres range between ≥20 and ≥1280 and are above the ≥8 threshold for an accepted correlate of protective immunity against *N. meningitidis*[35] (Table 2).

## Discussion

During infection pathogens present our immune system with an assortment of surface-exposed, lipid anchored, and integral membrane proteins, all of which can be used as components in subunit vaccines. While lipoproteins are easily engineered for recombinant expression (by removal of their lipid anchor), integral membrane proteins present several challenges for vaccine development. Recombinant forms of integral membrane proteins are often poorly expressed and their native conformations may be compromised during purification, potentially reducing their ability to elicit immune responses against conformational epitopes, such as those found in surface loops[10,42]. Furthermore, immunisation with integral membrane proteins can generate irrelevant immune responses, which are directed towards epitopes that are masked by the bacterial outer membrane[43]. To circumvent these issues, structure-based design was used to develop ChAs. We selected a key surface-exposed epitope (VR2) from the integral membrane protein PorA and inserted it into the immunogenic scaffold of the lipoprotein fHbp. We show that multivalent ChAs generate immune responses against two key surface antigens that can elicit protective immunity, providing proof-in principle that immunogenic epitopes from integral membrane proteins can be introduced into a soluble molecular scaffold to create functional antigens, an approach that could be applied to other antigens.

Combining two antigens within a single recombinant ChA could diminish the immunogenicity of one or both antigens. fHbp is highly immunogenic in all ChAs, inducing bactericidal antibody responses in excess of those correlated with protection[35]. We found PorA VR2 could induce antibody responses when displayed away from its native environment in the outer membrane. While all ChAs induced antibodies that detected PorA on the surface of *N. meningitidis*, not all induced bactericidal PorA antibodies. We observed a mixed bactericidal PorA response, which depended on the PorA VR2 peptide, the position of PorA VR2 in the fHbp scaffold and the adjuvant used for immunisation, indicating that several empirically determined parameters are crucial for obtaining bactericidal responses.

Previous studies found linear VR2 P1.16 peptides failed to elicit antibodies that recognised native PorA, while cyclic VR2 P1.16 peptides, with identical residues fixed in a β-turn, elicited PorA-specific bactericidal antibodies[40,44–47]. Structural observations using β-turn peptide mimics demonstrated that cyclised peptides, which mimic their native antigens, are able to induce functional immune responses[40,45,48]. In ChAs, the VR2 N- and C-termini are effectively locked into a 'biological' β-turn peptide mimic by neighbouring fHbp β-strands, as confirmed by our atomic structures. In previous work, cyclic peptides were coupled to carrier proteins and used with adjuvants not licensed for human use[40,44–46]. In the resulting SBAs, titres were only observed with antisera from some mice[45], similar to our findings with ChAs and licensed adjuvants.

Alum allows extended antigen presentation and stimulates T-helper (Th)-2 responses, predominantly producing IgG1 and IgE[49], while MPLA typically enhances Th1 responses, inducing IgG2a, IgG2b, and IgG3[50]. Murine IgG2a has a greater ability than murine IgG1 to activate the classical pathway[51]. Consistent with this, ChAs administered with MPLA elicited significantly higher fHbp-specific IgG2a responses and thus higher α-fHbp SBA titres than ChAs administered with alum. Of note, antisera raised against fHbp$^{V1.1}$:PorA$^{309}$/P1.16/MPLA had PorA-dependent SBA titre and contained significant levels of α-PorA IgG2a and IgM.

fHbp and PorA are both antigenically variable. To improve vaccine coverage, we used the comprehensive epidemiology data available for UK meningococcal isolates[41] to generate ChAs composed of the most prevalent fHbp and PorA antigens. This maximises vaccine coverage as ChA composition mirrors the prevalent fHbp and PorA antigens circulating within a given geographical area. For example, a vaccine composed of the three most common fHbp peptides from each variant group (fHbp peptides V1.4, V2.19 and V3.45) with a single PorA VR2 insertion (PorA VR2 P1.4, P1.9 and P1.14) would give exact sequence coverage against 57% of serogroup B strains circulating in the UK in 2016; this compares favourably with currently licensed meningococcal serogroup B vaccines Bexsero (36%) and Trumenba® (4.8%)[33,34,41].

In summary, using structure-based design, we generated ChAs that retain epitopes of fHbp and PorA and generate immune responses against both antigens, demonstrating that a soluble antigen can be exploited as a scaffold to display epitopes from an integral membrane protein. Our work provides proof-in-principle for bacterial vaccine design employing structure-led protein engineering previously used in viral proteins to graft functional motifs onto unrelated protein scaffolds[52–55], or β-hairpin peptide mimetics[56,57] to develop novel conformationally restricted antigens.

## Methods

**Bacterial strains and growth.** The bacterial strains used in this study are shown in Supplementary Table 2 and Supplementary Table 3. *N. meningitidis* was grown in the presence of 5% $CO_2$ at 37 °C on Brain Heart Infusion (BHI, Oxoid, Basingstoke, United Kingdom) plates with 5% (v/v) horse serum (Oxoid) at 37 °C. *E. coli* was

grown on Luria-Bertani (LB) agar plates or LB liquid at 37 °C supplemented with 100 µg µl$^{-1}$ carbenicillin.

**Expression and purification of ChAs.** N-terminally truncated fHbp V1.1 was amplified from MC58 genomic DNA using primers fHbp F1 and fHbp R4 (Supplementary Table 4). PorA VR2 P1.16 (YYTKDTNNNLTLV) was introduced into one of six positions in *fhbp* by overlap PCR using the primers in Supplementary Table 4. Four VR2 loops P1.4 (HVVVNNKVATHVP), P1.9 (YVDEQSKYHA), P1.10_1 (HFVQNKQNQPPTLVP), P1.14 (YVDEKKMVHA) and P1.15 (HYTRQNNADVFVP) were introduced into a single position in *fHbp* by overlap PCR using the primers in Supplementary Table 4. PCR products were digested with *NdeI* and *XhoI* (NEB) and then ligated into pET21b (Novagen); constructs were confirmed by sequencing. Protein expression was performed in *E. coli* strain B834. Expression cultures were incubated at 37 °C, upon reaching an OD$_{600}$ of ~0.8, protein expression was induced with 1 mM IPTG. Cultures were harvested after overnight expression at 37 °C. Bacteria were resuspended in Buffer A (50 mM Na-phosphate pH 8.0, 300 mM NaCl, 30 mM Imidazole) and purified by Nickel affinity chromatography (His-trap FF Crude; GE Healthcare). Columns were washed with 25 column volumes (CV) of Buffer A, then 20 CV 80:20 Buffer A:Buffer B (50 mM Na-phosphate pH 8.0, 300 mM NaCl, 300 mM Imidazole); elution was performed with 10 CV 40:60 Buffer A:Buffer B. Proteins were dialysed overnight at 4 °C into 50 mM Na-Acetate pH 5.5 buffer and further purification was achieved by ion exchange chromatography (HiTrapSP HP; GE Healthcare) at room temperature with a 0–1 M NaCl gradient in 50 mM Na-Acetate pH 5.5 buffer, followed by gel filtration using a HiLoad 16/600 Superdex 75 pg (GE Healthcare) column equilibrated with PBS (Oxoid).

**Generation of *N. meningitidis* strains.** Deletion of *porA* was performed by replacing the open reading frame with a tetracycline resistance cassette. Briefly, ~500 bp of the up- and downstream regions of the *porA* locus flanking a tetracycline resistance cassette were generated by PCR using primers PorA KO: F1, R1, F2, R2 (Supplementary Table 4) and the mega-primer method[58]. The PCR product was transformed into *N. meningitidis* H44/76 and H44/76∆*fHbp* as described previously[59]. Genomic DNA was obtained (Wizard® genomic DNA purification kit; Promega) from the resulting strains and both mutations were backcrossed into the WT H44/76 background using genomic DNA[59].

**Western blot analyses.** Western blots of purified proteins (0.5 µg) or 10 µl of *N. meningitidis* cell lysate[60] were probed with one of the following monoclonal antibodies or polyclonal sera: PorA P1.4 mAb (NIBSC cat: 02/148, diluted 1 in 500), PorA P1.9 mAb (NIBSC cat: 05/190, diluted 1 in 250), PorA P1.14 mAb (NIBSC cat: 03/142, diluted 1 in 500), PorA P1.15 mAb (NIBSC cat: 02/144, diluted 1 in 1,000), PorA P1.16 mAb (NIBSC cat: 01/538, diluted 1 in 1,000), pAb to fHbp V1.1[60] (diluted 1 in 1000) or polyclonal sera from mice immunised with a ChA (diluted 1 in 500). Following incubation with anti-mouse HRP-conjugated secondary antibodies (diluted 1 in 10,000), membranes were visualised via ECL (GE Healthcare) on a LAS-4000 (FujiFilm).

**Generation of immune sera.** Immunisations were performed with each ChA and PBS controls, using alum or MPLA as the adjuvant. Alum immunisations were prepared by incubating 20 µg ChA or PBS, 2% Alhydrogel (Invivogen), 10 mM Histidine-HCl pH 6.5 and 155 mM NaCl overnight at 4 °C on an end-over-end rocker. For MPLA immunisations, lyophilised MPLA (Invivogen) was resuspended in sterile H$_2$O by incubating for 5 min in a sonicating water bath. Ten micrograms of MPLA were mixed at room temperature with 20 µg ChA or PBS, 10 mM Histidine-HCl pH 6.5 and 155 mM NaCl. Groups of eight female CD1 mice (~6 weeks old, Charles Rivers, Margate) were immunised by the intraperitoneal route on days 0, 21 and 35. Four mice were kept in each cage, which were individually vented. Sera was obtained on day 49 following cardiac puncture under terminal anaesthesia. All procedures were conducted in accordance with UK Home Office guidelines. Ethical approval was obtained by the Central Committee on Animal Care and Ethical Review at the University of Oxford. All sera were stored at −80 °C until required; once defrosted, sera were stored at 4 °C.

**Serum bactericidal assays.** SBAs were performed as previously described[60], with the following modifications. *N. meningitidis* was suspended in Dulbecco's PBS with cations (Gibco) supplemented with 0.1% glucose (DPBS-G) to a final concentration of $1.25 \times 10^4$ CFU ml$^{-1}$. Baby rabbit complement (Cedar lane, lot #15027680) was diluted with DPBS-G to a final dilution of 1/10. Serum, pooled or from individual mice, was heat inactivated for 1 h at 56 °C and added to the wells in a serial two-fold dilution, starting with a dilution of 1/5 or higher. Control wells contained no serum or no complement. Following static incubation for 1 h at 37 °C in the presence of 5% CO$_2$, 10 µl from each well was plated onto BHI plates in triplicate and colonies from surviving bacteria counted. Bactericidal activity is expressed as the dilution of serum required to kill≥50% of bacteria in assays containing both complement and serum in comparison with control assays containing serum or complement alone. SBAs using pooled sera were repeated three times, and assays using sera from individual mice were repeated twice. SBA titres were input into GraphPad Prism and statistical analyses comparing titres obtained from alum

immunisations with titres obtained from MPLA immunisations were performed using two-way ANOVA (statistical significance of $p \le 0.05$) and Dunnett's method of multiple comparisons.

**Surface plasmon resonance.** SPR was performed using a Biacore 3000 (GE Healthcare). Recombinant ChAs were dissolved in 50 mM sodium acetate pH 4.5 and immobilised on a CM5 sensor chip (GE Healthcare). Increasing concentrations of CFH$_{6/7}$ (1–16 nM) were injected over the flow channels at 40 µl min$^{-1}$). Dissociation was allowed for 300 s. BIAevaluation software was used to calculate the $K_d$.

**Structural biology.** ChAs in PBS were concentrated to 7 mg ml$^{-1}$ and screened for crystal formation via the sitting drop method at a 0.4:0.6 ratio of protein to mother liquor. Crystals of chimeras fHbp$^{V1.4}$:PorA$^{151/P1.16}$, fHbp$^{V1.1}$:PorA$^{294/P1.16}$, fHbp$^{V1.1}$:PorA$^{309/P1.16}$ and fHbp$^{V1.4}$:PorA$^{309/P1.16}$ grew under the respective conditions: 2.0 M ammonium sulphate, 0.15 M sodium citrate pH 5.5; 20% (w/v) PEG4000, 0.3 M ammonium sulphate; 0.01 M zinc chloride, 0.1 M sodium acetate, 20% (w/v) PEG6000; and 0.2 M potassium formate, 20% (w/v) PEG3350 respectively. Crystals were transferred to a cryoprotectant solution comprised of mother liquor and 30% ethylene glycol. Data were collected at Diamond light source and integrated/scaled using the Xia2 programme. Molecular replacement was carried out using Phaser from the CCP4 package[61], with the fHbp V1.1 structure used as a search model (PDB 2W80). Refmac5[62], Coot[63] and Phenix[64–67] were utilised for rebuilding and refinement, with structural validation performed in Molprobity[68]. Refinement statistics for each structure are detailed in Supplementary Table 5.

**Differential scanning calorimetry.** Purified ChAs (20 µM) in PBS were subjected to a 20 to 120 °C temperature gradient on a Malvern VP Capillary DSC. Melting temperature ($T_m$) is recorded for the fHbp N-terminal and C-terminal β-barrels (Table 1, Supplementary Figure 1).

**Flow cytometry.** *N. meningitidis* strains H44/76, H44/76∆*fHbp*, H44/76∆*porA* and H44/76∆*fHbp*∆*porA* ($1 \times 10^9$ cells) were fixed in 3% paraformaldehyde (PFA) for 1 h. PFA was removed by washing cells three times with PBS. To measure binding of serum antibodies to fHbp and PorA, $1 \times 10^8$ cells were incubated for 1 h at 4 °C, shaking at 1250 rpm, with sera diluted 1/50 in PBS. Cells were washed three times with PBS, then incubated for 30 min at 4 °C with shaking at 1250 rpm, with a secondary antibody (all ThermoFisher Scientific). Final dilutions of secondary antibodies were as follows: 5 µg ml$^{-1}$ for Alexa-488 conjugated anti-mouse IgAGM, and 2.5 µg ml$^{-1}$ for each of Alexa-488 conjugated anti-mouse IgM, IgG3 and IgG2b, and Alexa-647 conjugated anti-mouse IgG2a and IgG1. Cells were washed three times with PBS and then resuspended in PBS for analysis on a FACSCalibur (BD Biosciences). Data were imported into the FlowJo$^{TM}$ data analysis package and transformed to the Logicle scale. A uniform gate was applied to all data sets, based on bacteria visualised by forward and side-scatter, and the geometric mean of fluorescence calculated for FL1H (Alexa-488 signal) and FL4H (Alexa-647 signal). Each flow experiment was repeated three times, and the FL1H and FL4H geometric means were input into Graphpad Prism. Two-way ANOVA (statistical significance of $p \le 0.05$, using Dunnett's method of multiple comparison) was used to compare the Alexa-488 or Alexa-647 geometric means with control Alexa-488 or Alexa-647 geometric means. Control results were obtained by incubating PFA fixed cells with sera from mice immunised with PBS and Alum or MPLA adjuvants, and the corresponding Alexa-fluor-labelled secondary Ig. Two-way ANOVA multiple comparisons were performed with geometric means from sets of experiments using the same secondary antibody, *N. meningitidis* strain and adjuvant; the only variable factor was the ChA used for immunisation.

**Data availability.** Structural data that support the findings of this study have been deposited in RCSB Protein Data Bank with the primary accession codes 5NQP for fHbp$^{V1.4}$:PorA$^{151/P1.16}$; 5NQX for fHbp$^{V1.1}$:PorA$^{294/P1.16}$; 5NQZ for fHbp$^{V1.1}$:PorA$^{309/P1.16}$; and 5NQY for fHbp$^{V1.4}$:PorA$^{309/P1.16}$.

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

## Acknowledgements

We gratefully acknowledge financial support from Action Medical Research (Award number GN2205). We thank David Staunton for performing the DSC analysis, Sophie Andrews and Sasha Burgess for assisting with the production of expression vectors, Diamond and ESRF synchotrons for provision of beam time, and the Diamond-I02, Diamond-I04 and ESRF-ID29 beamline staff for help with beamline preparation and data collection. We gratefully thank the Meningococcal Reference Unit, Public Health England, for the meningococcal serogroup B disease isolates. Nicola Ruivo provided technical assistance.

## Author contributions

S.H. and I.J. contributed by designing and generating constructs, protein purification, performing immunisations, analysing immune responses, and experimental design. S.H. also helped with setting up crystallisation and structural analyses. R.M.E. contributed to experimental design. S.J. and S.M.L. contributed to structural studies, SPR and design of constructs, C.M.T. contributed to project organisation and experimental design. S.H. prepared the figures. S.H., C.M.T., S.J., I.J., R.M.E. and S.M.L. wrote the manuscript.

## Additional information

**Competing interests:** The authors declare that they have filed a patent on Chimeric antigens (ref GB1614687, PCT filing date: 31st August 2017).

