## [Peer Review File · Nature Communications]

Reviewers' comments:

Reviewer #1 (Remarks to the Author):

Bacterial meningitis is a deadly disease. Meningococcal vaccines are currently available to protect against five invasive meningococcal serogroups A, B, C, W and Y. Protein-based vaccines against serogroup B may not provide broad coverage against all B strains due to their antigenic diversity. This manuscript described the development of a chimeric antigens containing immunogens fHbp and PorA. The study used one of the prevalent fHbp variant as a molecular scaffold with PorA VR2 region inserted into six different beta-turn regions. These constructs are relatively stable and retain the architecture of the scaffold and PorA variable loop. These chimeric antigens have also proven to elicit immunogen specific response and bactericidal activities. The paper is well written and organized. The study design is creative and provides an alternative for developing protein-based vaccines.

Major comments

1. While fHbp is able to provide cross-protection against strains containing fHbp of the same variant/subfamily, protection provided by PorA is very sequence or strain specific, having one fHbp and PorA typing in a chimeric antigen will not cover all B strains worldwide. How do you propose to use this approach to improve vaccine coverage? If a mixture of different ChAs are included in a vaccine, how does this impact the capability of each ChA inducing bactericidal immunity? Comments on these questions in discussion would be helpful.
2. Have you checked if the conserved regions of PorA antigen can induce immune response other than the VR2, which is hyper-variable. Or other vaccine antigens that can potentially provide coverage for non-B strains? Seems that if this elegant approach is going to be used to improve the B vaccine coverage, regions within fHbp and PorA that can provide cross-protection against diverse strains are absolutely needed.
3. The data suggested the position in fHbp scaffold where the PorA VR2 was inserted really matters. Any lessons learned about how to choose these positions can be shared with the audiences?
4. Would it possible to insert different VR2 regions into one fHbp scaffold at different positions? I assume this has higher chance to change fHbp architecture. Just curious.

Minor comments

1. Line 15. Should be MenB instead of Menb
2. Line 32. Influenzae missed "e" at the end
3. Line 58 fHbp is a protein name, if want to use plural perhaps say fHbp peptides or subvariants instead of fHbps.
4. Line 59. It seems variant is used to refer a fHbp peptide and the subfamily. Suggest using different terms for peptide and subfamily. Literatures used subvariant for fHbp peptide, and variant or subfamily for a group of fHbp peptides.
5. Line 84. Out of all the loops PorA has, only VR2 is likely soluble when expressed as a separated peptide?
6. Line 104. Is the CFH binding to the ChAs comparable to the wild type fHbp?
7. Fig 2C/E. High reactivity was found in both fHbp and porA single mutant, is there an explanation for that?

Reviewer #2 (Remarks to the Author):

The manuscript from Hollingshead et al reports the design and characterization of chimeric antigens (ChAs) against serogroup B N. meningitidis. The antigens use factor H binding protein (fHBP) as a scaffold onto which the variable region 2 (VR2) loop of PorA is grafted. The grafting uses structural information of fHBP and the VR2 loop so that the VR2 loop is inserted into beta-turns on fHBP, allowing the immunogenic beta-hairpin conformation of VR2 to be preserved. Mice

are immunized with several different ChA variants using alum or MPLA as adjuvants, and the sera are tested for binding reactivity and bactericidal activity. In general, the ChAs elicit sera that bind well to cells expressing fHBp. The sera do bind to cells expressing PorA, but the reactivity is weaker and more variable. A similar trend is observed for the bactericidal activity of the sera, with robust SBA titers against cells expressing fHBP, but very poor or no SBA titers against cells expressing PorA (and not fHBp). Only certain mice from some immunization regimens had any measurable PorA-specific SBA titers. The authors also determined crystal structures for 4 ChAs, which revealed that the fHBP structure was not substantially altered by the insertion of the VR2 loops, and that the VR2 loops adopted the immunogenic conformation regardless of the insertion point.

Overall, I found the manuscript interesting, and the experiments were performed well with the necessary controls. The conclusions are also supported by the data. The SBA titers elicited against the grafted PorA VR2 loop were disappointing though, suggesting that much additional work is needed to optimize such ChAs. But as a proof-of-principle study, the results are sufficient. The writing needs to be improved for clarity, and would benefit from a copy-editing service, but the figures are generally clear and well organized.

Concerns to be addressed:

1. The authors claim in the discussion that this study represents a proof of principle that immunogenic epitopes from integral membrane proteins can be introduced into soluble molecular scaffolds to create ChAs. However, such epitope grafting has already been demonstrated for numerous viral glycoproteins, including HIV-1 Env and RSV F. It would be helpful if the authors acknowledged this previous work from Bill Schief, Peter Kwong, and others, and discussed how the ChAs described here are similar or different to previously described approaches.
2. Biacore traces and DSC curves for all ChAs need to be included in one or more supplemental figures, otherwise the fits and quality of the data in the tables cannot be assessed.

Minor concerns:

1. Table 1: Why is "Cp (kcal mole⁻¹ C⁻¹)" at the top of the Table? The melting temperatures and Kds are provided, not specific heats.
2. Figure 4F: What do the blue arrowheads indicate on the Western blot? They are not described in the legend. Is there a reason why some bands are shifted or exist as a doublet?
3. Table S5: Bond angle rmsds only need to be reported to two decimal points.

Reviewer #1 (Remarks to the Author):

Bacterial meningitis is a deadly disease. Meningococcal vaccines are currently available to protect against five invasive meningococcal serogroups A, B, C, W and Y. Protein-based vaccines against serogroup B may not provide broad coverage against all B strains due to their antigenic diversity. This manuscript described the development of a chimeric antigens containing immunogens fHbp and PorA. The study used one of the prevalent fHbp variant as a molecular scaffold with PorA VR2 region inserted into six different beta-turn regions. These constructs are relatively stable and retain the architecture of the scaffold and PorA variable loop. These chimeric antigens have also proven to elicit immunogen specific response and bactericidal activities. The paper is well written and organized. The study design is creative and provides an alternative for developing protein-based vaccines.

We are grateful for the reviewer's positive comments about our work.

Major comments

1. While fHbp is able to provide cross-protection against strains containing fHbp of the same variant/subfamily, protection provided by PorA is very sequence or strain specific, having one fHbp and PorA typing in a chimeric antigen will not cover all B strains worldwide. How do you propose to use this approach to improve vaccine coverage? If a mixture of different ChAs are included in a vaccine, how does this impact the capability of each ChA inducing bactericidal immunity? Comments on these questions in discussion would be helpful.

We have altered the Discussion as suggested and it now includes (line 271):

“fHbp and PorA are both antigenically variable. To improve vaccine coverage, we used the comprehensive epidemiology data available for UK meningococcal isolates⁵⁴ to generate ChAs composed of the most prevalent fHbp and PorA antigens. Thus, vaccine coverage can be maximised by tailoring ChA composition to match the prevalent fHbp and PorA antigens circulating within a given geographical region. For example, a vaccine composed of the three most common fHbp peptides from each variant group (fHbp peptides V1.4, V2.19 and V3.45) with a single PorA VR2 insertion (PorA VR2 P1.4, P1.9 and P1.14) would give exact sequence coverage against 57% of serogroup B strains that were circulating in the UK in 2016. This compares favourably with the currently licensed meningococcal serogroup B vaccines^{42,43}, Bexsero and Trumenba, which have exact sequence matches to 36% and 4.8% of circulating isolates, respectively.”

2.a) Have you checked if the conserved regions of PorA antigen can induce immune response other than the VR2, which is hyper-variable.

PorA is predicted to contain eight surface exposed loops, three of which are hyper-variable and known as VR1 (loop 1), VR2 (loop 4) and VR3 (loop 5). Of these, the VR2 elicits most immune response. The remaining loops are more conserved; however, antibodies against loops 2, 3, 6 and 7 do not bind intact bacteria or PorA in outer membrane vesicles (Van der Ley et al, *Infection and Immunity* 59, 2963-2971, 1991). Therefore, it is unlikely these loops are exposed to the immune system *in vivo*, and would not induce protective immune responses.

b) Or other vaccine antigens that can potentially provide coverage for non-B strains? Seems that if this elegant approach is going to be used to improve the B vaccine coverage, regions within fHbp and PorA that can provide cross-protection against diverse strains are absolutely needed.

Our aim is to produce ChAs directed against *Neisseria meningitidis* serogroup B strains, which could be used with the excellent glycoconjugate vaccines already available against the other disease causing serogroups (*i.e.* A, C, W, X and Y). Whilst we are interested in testing loops from other vaccine antigens, such as FetA and ZnuD, to improve coverage, this work is beyond the scope of our current study. In line with the reviewer's comments, we have included this point in the Discussion (line 228).

3. *The data suggested the position in fHbp scaffold where the PorA VR2 was inserted really matters. Any lessons learned about how to choose these positions can be shared with the audiences?*

The reviewer is correct. The positions in fHbp were chosen as they form loops that extend away from the β -barrels of the protein. However, the optimal position of PorA loops has to be determined empirically; different PorA VR2s have differing expression levels in each of position in different fHbps. For example, we get expression of fHbp V3.45 containing any of the PorA VR2s P1.2, P1.4 or P1.9 in position 151, but observe little or no expression of these PorA VR2s in the corresponding positions in fHbp V1.4. We have added text to the discussion to highlight this point (line 239).

4. *Would it possible to insert different VR2 regions into one fHbp scaffold at different positions? I assume this has higher chance to change fHbp architecture. Just curious.*

To date, we have tried inserting different PorA VR2s into two different positions (amino acids 151 and 267) in fHbp V1.1, however, very little expression was observed using the construct. We are currently investigating other VR2s and other combinations of positions in fHbp.

Minor comments:

1. *Line 15. Should be MenB instead of Menb*

Thank you. We have corrected this.

2. *Line 32. Influenzae missed "e" at the end*

Thank you. We have corrected this.

3. *Line 58 fHbp is a protein name, if want to use plural perhaps say fHbp peptides or subvariants instead of fHbps.*

We have modified "fHbps" to "fHbp peptides" throughout the manuscript.

4. *Line 59. It seems variant is used to refer a fHbp peptide and the subfamily. Suggest using different terms for peptide and subfamily. Literatures used subvariant for fHbp peptide, and variant or subfamily for a group of fHbp peptides.*

We have modified this to "Trumenba is solely composed of two fHbp peptides".

5. *Line 84. Out of all the loops PorA has, only VR2 is likely soluble when expressed as a separated peptide?*

Given the hydrophilic nature of the other surface exposed PorA loops, it is possible these will be soluble when expressed as a separated peptide. However, we have only tested different VR2 loops as, all studies investigating immune responses generated by commercial outer membrane vesicle vaccines demonstrate the majority of protective immunity is generated by the VR2 loop, with smaller contributions from VR1 and negligible responses from the other loops.

6. *Line 104. Is the CFH binding to the ChAs comparable to the wild type fHbp?*

CFH binding to the ChAs is comparable to wild-type fHbp for four ChAs; we have altered the text to "Similar to wild-type fHbp, most ChAs bind CFH at high affinity, indicating the fHbp scaffold retains its function. The exceptions were fHbp^{V1.1}:PorA^{183/P1.16}, to which there was no detectable CFH binding, and fHbp^{V1.1}:PorA^{267/P1.16}, to which CFH binding was reduced by approximately eight fold."

7. *Fig 2C/E. High reactivity was found in both fHbp and porA single mutant, is there an explanation for that?*

This is due to background from the polyclonal antisera, and can be observed in the histograms of the flow cytometry data (Figure S4) from which the geometric means were calculated.

Reviewer #2 (Remarks to the Author):

The manuscript from Hollingshead et al reports the design and characterization of chimeric antigens (ChAs) against serogroup B N. meningitidis. The antigens use factor H binding protein (fHbp) as a scaffold onto which the variable region 2 (VR2) loop of PorA is grafted. The grafting uses structural information of fHbp and the VR2 loop so that the VR2 loop is inserted into beta-turns on fHbp, allowing the immunogenic beta-hairpin

conformation of VR2 to be preserved. Mice are immunized with several different ChA variants using alum or MPLA as adjuvants, and the sera are tested for binding reactivity and bactericidal activity. In general, the ChAs elicit sera that bind well to cells expressing fHBp. The sera do bind to cells expressing PorA, but the reactivity is weaker and more variable. A similar trend is observed for the bactericidal activity of the sera, with robust SBA titers against cells expressing fHBp, but very poor or no SBA titers against cells expressing PorA (and not fHBp). Only certain mice from some immunization regimens had any measurable PorA-specific SBA titers. The authors also determined crystal structures for 4 ChAs, which revealed that the fHBp structure was not substantially altered by the insertion of the VR2 loops, and that the VR2 loops adopted the immunogenic conformation regardless of the insertion point.

Overall, I found the manuscript interesting, and the experiments were performed well with the necessary controls. The conclusions are also supported by the data. The SBA titers elicited against the grafted PorA VR2 loop were disappointing though, suggesting that much additional work is needed to optimize such ChAs. But as a proof-of-principle study, the results are sufficient. The writing needs to be improved for clarity, and would benefit from a copy-editing service, but the figures are generally clear and well organized.

We are grateful for the reviewer's positive comments about our work. We have modified the text of the article in several places to improve its clarity.

Concerns to be addressed:

1. The authors claim in the discussion that this study represents a proof of principle that immunogenic epitopes from integral membrane proteins can be introduced into soluble molecular scaffolds to create ChAs. However, such epitope grafting has already been demonstrated for numerous viral glycoproteins, including HIV-1 Env and RSV F. It would be helpful if the authors acknowledged this previous work from Bill Schief, Peter Kwong, and others, and discussed how the ChAs described here are similar or different to previously described approaches. We have updated our Discussion to acknowledge the work of other authors on β -peptide mimetics and epitope grafting.

Line 245 "Structural observations using β -turn peptide mimetics demonstrate that cyclised peptides which mimic their native antigens are able to induce functional immune responses^{52,53,61}."

Line 288 "In summary, using structure-based design, we generated ChAs that retain epitopes of fHbp and PorA and generate immune responses against both antigens, demonstrating that a soluble antigen can be exploited as a scaffold to display epitopes from an integral membrane protein. Our work provides proof-in-principle in bacterial vaccine design of structure-led protein engineering previously used in viral proteins to graft functional motifs onto unrelated protein scaffolds⁷⁰⁻⁷³, or β -hairpin peptide mimetics⁷⁴⁻⁷⁷ to develop novel conformationally restricted antigens."

2. Biacore traces and DSC curves for all ChAs need to be included in one or more supplemental figures, otherwise the fits and quality of the data in the tables cannot be assessed.

In line with the reviewer's suggestions, the DSC curves are now included as Figure S1, and the Biacore traces have been added as Figure S3.

Minor concerns:

1. Table 1: Why is " C_p (kcal mole⁻¹ C⁻¹)" at the top of the Table? The melting temperatures and Kds are provided, not specific heats.

We have updated the Table to include: Melting Temperature (°C)

2. Figure 4F: What do the blue arrowheads indicate on the Western blot? They are not described in the legend. Is there a reason why some bands are shifted or exist as a doublet?

We apologise for this omission. The blue arrows indicate PorA specific bands, as there is a non-specific band observed in all isolates, which migrates just below PorA on the Western

blots (this non-specific band possibly represents the double band the reviewer refers to). We have added this clarification to the figure legend.

3. Table S5: Bond angle rmsds only need to be reported to two decimal points.

We have altered the Table as suggested.

REVIEWERS' COMMENTS:

Reviewer #1 (Remarks to the Author):

The revised version is much clear and improved. Thank you for addressing my questions.

Reviewer #2 (Remarks to the Author):

The authors have satisfactorily addressed my previous concerns and comments, and the inclusion of the SPR and DSC data is appreciated.